# AIRE is induced in oral squamous cell carcinoma and promotes cancer gene expression

Chi Thi Kim Nguyen[1¤a], Wanlada Sawangarun[1¤b], Masita Mandasari[1¤c], Kei-ichi Morita[2,3], Hiroyuki Harada[4], Kou Kayamori[1], Akira Yamaguchi[1,5], Kei Sakamoto[1]*

**1** Department of Oral Pathology, Graduate School of Medical and Dental Sciences, Tokyo Medical and Dental University, Tokyo, Japan, **2** Department of Maxillofacial Surgery, Graduate School of Medical and Dental Sciences, Tokyo Medical and Dental University, Tokyo, Japan, **3** Bioresource Research Center, Tokyo Medical and Dental University, Tokyo, Japan, **4** Department of Oral and Maxillofacial Surgery, Graduate School of Medical and Dental Sciences, Tokyo Medical and Dental University, Tokyo, Japan, **5** Oral Health Science Center, Tokyo Dental College, Tokyo, Japan

¤a Current address: Oral Pathology Department, Faculty of Dentistry, University of Medicine and Pharmacy, Hochiminh City, Vietnam
¤b Current address: Department of Oral Diagnosis, Faculty of Dentistry, Naresuan University, Phitsanulok, Thailand
¤c Current address: Oral Medicine Department, Faculty of Dentistry, Universitas Indonesia, Jakarta, Indonesia
* s-kei.mpa@tmd.ac.jp

**Data Availability Statement:** All relevant data are within the paper and its Supporting Information files.

## Abstract

Autoimmune regulator (AIRE) is a transcriptional regulator that is primarily expressed in medullary epithelial cells, where it induces tissue-specific antigen expression. Under pathological conditions, AIRE expression is induced in epidermal cells and promotes skin tumor development. This study aimed to clarify the role of AIRE in the pathogenesis of oral squamous cell carcinoma (OSCC). AIRE expression was evaluated in six OSCC cell lines and in OSCC tissue specimens. Expression of STAT1, ICAM1, CXCL10, CXCL11, and MMP9 was elevated in 293A cells stably expressing AIRE, and conversely, was decreased in AIRE-knockout HSC3 OSCC cells when compared to the respective controls. Upregulation of STAT1, and ICAM in OSCC cells was confirmed in tissue specimens by immunohistochemistry. We provide evidence that AIRE exerts transcriptional control in cooperation with ETS1. Expression of STAT1, ICAM1, CXCL10, CXCL11, and MMP9 was increased in 293A cells upon *Ets1* transfection, and coexpression of AIRE further increased the expression of these proteins. AIRE coprecipitated with ETS1 in a modified immunoprecipitation assay using formaldehyde crosslinking. Chromatin immunoprecipitation and quantitative PCR analysis revealed that promoter fragments of STAT1, ICAM1, CXCL10, and MMP9 were enriched in the AIRE precipitates. These results indicate that AIRE is induced in OSCC and supports cancer-related gene expression in cooperation with ETS1. This is a novel function of AIRE in extrathymic tissues under the pathological condition.

**Funding:** This work was supported by JSPS (Japan Society for the Promotion of Science, https://www.jsps.go.jp/english/index.html) KAKENHI [JP25462848 and JP16K11438] to K.S.. This work was supported in parts by a grant for Private University Branding Project to Tokyo Dental College from MEXT (Ministry of Education, Culture, Sports, Science and Technology, Japan, http://www.mext.go.jp/). The funders had no role in study design, data collection and analysis, decision to publish, or preparation of the manuscript.

**Competing interests:** The authors have declared that no competing interests exist.

## Introduction

The burden of oral cancer has significantly increased in many parts of the world, causing more than 145.000 death in 2012 worldwide [1]. Despite the advances in modern medicine, oral cancer can have devastating effects on critical life functions. About 90 percent of oral cancers are squamous cell carcinomas (OSCCs) that arise from keratinocytes of the oral mucosa.

Cancer development requires the acquisition of several properties, such as unlimited proliferation, vascularization, and invasion [2]. Aberrant growth control in cancer cells is the consequence of accumulated disorders in multiple cell-regulatory systems. Molecular analysis of OSCCs has uncovered genetic and epigenetic alterations in several distinct driver pathways, including mitogenic signaling and cell-cycle regulatory pathways, which lead to continual and unregulated proliferation of cancer cells. In addition, cancer cells recruit surrounding non-transformed cells, such as stromal cells and inflammatory cells, to create a tumor microenvironment that fosters tumor growth and invasion. Under the influence of interaction with the tumor microenvironment, cancer cells express a unique group of proteins that are absent or expressed at very low levels in normal cells. Differentially expressed proteins in OSCC compared to normal keratinocytes include various cytokines, chemokines (for example, CXCLs), extracellular matrix proteins, matrix remodeling enzymes (for example, matrix metalloproteases (MMPs)), cell adhesion molecules, and cytoskeletal proteins [3,4]. Among these factors that characterize cancer cells, we have been interested in keratin 17 (KRT17) because it is consistently and strongly induced in OSCC, and because of its unique functions. Keratins are a family of intermediate filaments that are indispensable for the structural stability of epithelial cells. KRT17 is regarded a "stress-response keratin" that is not expressed or at a very low level in normal squamous epithelium and is induced under pathological conditions, such as wound healing, inflammation, and cancer [5]. Interestingly, KRT17 has been recently reported to interact with AIRE [6]. *AIRE* was first identified as a causative gene for autoimmune polyendocrinopathy-candidiasis-ectodermal dystrophy syndrome, which gives rise to multiple endocrine disorders, chronic mucocutaneous candidiasis, and various ectodermal defects.

AIRE consists of four domains, including a caspase recruitment domain/homogeneously staining region (CARD/HSR) domain that drives nuclear localization and oligomerization, an Sp100, AIRE, NucP41/75, DEAF1 (SAND) domain, and plant homeodomain 1 (PHD1) and PHD2 domains. The closest homolog of AIRE is Sp100, which shares the CARD, SAND, and PHD domains [7]. AIRE localizes in the nucleus in discrete, punctate structures that resemble promyelocytic leukemia nuclear bodies, which contain Sp100 family members. The nuclear localization and the presence of domains that are shared by transcription factors suggested a role for AIRE in the regulation of gene expression, and indeed, numerous *in-vitro* studies have demonstrated that AIRE functions as a transcriptional activator [7]. AIRE is expressed in medullary thymic epithelial cells (mTECs), in which it induces ectopic expression of peripheral tissue-specific antigens, thereby facilitating the elimination of self-reactive T cells. AIRE expression has been also reported in extrathymic tissues, including epidermal keratinocytes [8–11]. In the mouse epidermis, Aire is induced by inflammatory or tumorigenic stimuli, and stimulates the transcription of several chemokine genes, such as *Cxcl10* and *Cxcl11*, and a gene encoding matrix-degrading enzyme, *Mmp9*, in keratinocytes [6]. These results suggest a novel role of AIRE in extrathymic epithelial tissues under pathological conditions; however, the significance of AIRE in other cancers remains to be elucidated.

The above findings inspired us to evaluate whether AIRE participates in the pathogenesis of OSCC. To this end, we first assessed the expression of AIRE in OSCC. Then, we explored the role of AIRE in the regulation of other genes associated with OSCC development.

## Materials and methods

### Chemically induced mouse model of tongue and esophageal cancer

Ten CD-1 mice were obtained from Sankyo Labo Service Co. (Tokyo, Japan) and were housed in groups of less than five in plastic cages in a temperature-controlled room with 12-h light/ 12-h dark cycles, and were provided free access to tap water and feed. To establish the cancer model, 4-nitroquinoline-1-oxide (4-NQO, MPN 147–03421, Lot AWQ5167, FUJIFILM Wako Chemical, Miyazaki, Japan) was added to the drinking water at a concentration of 100 μg/ml and administered to 6-week-old mice for 8 weeks. After the administration period, normal drinking water was given. The health of the animals was monitored daily. Behavioral changes were used as criteria to determine a humane endpoint. None of the mice reached the endpoint before the day of sacrifice. Mice were euthanized by cervical dislocation at 30 weeks of age, and tongues and esophagi were harvested. The animal studies were reviewed and approved by the institutional animal care and use committee of Tokyo Medical and Dental University (registration numbers: 0150213A and 0160316A).

### Histology and immunohistochemistry

The tissues were fixed in 10% neutral buffered formalin for 24–48 h and embedded in paraffin. Four micrometer-thick tissue sections were cut and stained with hematoxylin and eosin. For immunohistochemistry, the tissue sections were incubated in antigen retrieval buffer (10 mM Tris (pH 9.0), 1 mM EDTA) at 120˚C for 15 min. Endogenous peroxidase activity was quenched by 3% $H_2O_2$ for 10 min. The sections were incubated with primary antibody diluted 1:500 in TBST (10 mM Tris (pH 7.4), 150 mM NaCl, 0.1% Tween 20) at 4˚C overnight. After washing, the sections were incubated with a secondary antibody at room temperature for 1 h. 3,3′-Diaminobenzidine (DAB) was used for chromogenic detection. Primary antibodies used in this study were anti-AIRE (C-2, Santa Cruz Biotechnology, Santa Cruz, CA, USA) for detection of human AIRE, anti-AIRE (M300, Santa Cruz Biotechnology) for detection of mouse AIRE, anti-GAPDH (D16H11, Cell Signaling Technology, Danvers, MA, USA), anti-STAT1 alpha (EPYR2154, Abcam, Cambridge, UK), anti-STAT1 phospho S727 (EPR3146, Abcam), anti-ICAM1 (EP1442Y, Abcam), anti-ETS1 (D8O8A, Cell Signaling Technology), anti-keratin 14 (LL002, Abcam), anti-Flag M2 (Sigma-Aldrich, St. Louis, MO, USA), and anti-HA (12CA5, Roche Diagnostics, Basel, Switzerland). Secondary antibodies used in this study were Envision FLEX+ (Dako, Glostrup, Denmark), HRP-donkey anti-rabbit IgG (Thermo Fisher Scientific, Waltham, MA, USA), and HRP-rabbit anti-mouse IgG, Alexa Fluor 488 goat anti-rabbit IgG, Alexa Fluor 488 goat anti-mouse IgG, Alexa Fluor 594 goat anti-rabbit IgG, and Alexa Fluor 594 goat anti-mouse IgG (Thermo Fisher Scientific).

### Cell culture

Ca9-22 (derived from OSCC of the gingiva) and HSC-3 (derived from OSCC of the tongue, referred to as HSC3 hereafter) cell lines were obtained from the RIKEN Bioresource Center (Tsukuba, Japan). HSC5 (derived from SCC of the skin) and Ho-1-N-1 (derived from OSCC of the buccal mucosa, referred to as HO1N1 hereafter) were obtained from the Japanese Collection of Research Bioresources (Osaka, Japan). BHY (derived from OSCC of the gingiva) [12] was provided by Dr. Masato Okamoto (Tsurumi University). SAS (derived from OSCC of the tongue) [13] and HSC-4 (derived from OSCC of the tongue, referred to as HSC4 hereafter) [14] were provided by Dr. Masao Saito (Yamanashi University). 293A was purchased from Thermo Fisher Scientific Co. These cells were maintained in Dulbecco's modified Eagle's medium/nutrient mixture F-12 (Sigma-Aldrich) with 10% fetal bovine serum at 37˚C in the

presence of 5% $CO_2$. Primary human keratinocytes isolated from neonatal foreskins were obtained from Kurabo Co. (Osaka, Japan) and were maintained in HuMedia-KG2 (Kurabo) at 37°C in the presence of 5% $CO_2$.

## Plasmid constructs

A pET32a plasmid containing human *AIRE* cDNA was provided by Dr. Yoshitaka Yamaguchi (Keio University). The open reading frame of *AIRE* was recovered by PCR using this plasmid as a template and PrimeSTAR GXL DNA Polymerase (Takara, Shiga, Japan). The PCR product was digested with *Eco*R1 and inserted into pcDNA6-3xFlagN (in-house plasmid made using pcDNA6/V5-HisA (Thermo Fisher Scientific) as a backbone vector), or pAcGFP-C2 (Takara) to create N-terminally 3xFlag-tagged AIRE (*Flag-AIRE*), and GFP-tagged AIRE (*GFP-AIRE*), respectively. The plasmid construct targeting exon 2 of human *AIRE* (*AIREKO*) was created using pSpCas9(BB)-2A-GFP (a gift from Dr. Feng Zhang, #48138, Addgene, http://n2t.net/addgene:48138 ; RRID:Addgene_48138) as a backbone vector [15]. The guide sequence was GGAGCGCTATGGCCGGCTGC. pCMV-mFlagEts1 was a gift from Dr. Barbara Graves (# 86099, Addgene; http://n2t.net/addgene:86099; RRID:Addgene_86099) [16]. To remove the Flag tag, a fragment containing the coding region of mouse *Ets1* was amplified by PCR, digested with *Bam*H1 and *Spe*1, and inserted into pcDNA6/V5-HisA.

## Generation of AIRE-overexpressing and AIRE-knockout cells

Transfection was carried out using Polyethylenimine Max (Polysciences, Warrington, PA, USA). To establish AIRE-overexpressing clones, 293A cells were transfected with *Flag-AIRE* and selected on blasticidin (5 μg/ml), and clones were isolated from colonies. To establish knockout clones, HSC3 cells were transfected with *AIREKO*, GFP-positive cells were sorted using BD FACSAria II (BD Biosciences, San Jose, CA, USA) 72 h after transfection to compensate for the low transfection efficiency of HSC3 cells, and expanded through limiting dilution. Clones harboring a nonsense mutation were determined by PCR-direct sequencing using a BigDye Terminator v3.1 Cycle Sequencing Kit (Thermo Fisher Scientific).

## Immunofluorescence staining of cultured cells

Cells cultured in a chamber slide were fixed with methanol for 10 min. The cells were permeabilized with 0.5% Triton X-100 for 15 min, washed in PBS, incubated with Image-iT FX Signal Enhancer (Thermo Fisher Scientific) at room temperature for 30 min, and then incubated with primary antibodies (1:500 dilution) for 3 h, followed by incubation with secondary antibodies (1:500 dilution) for 1 h. Nuclei were counterstained with propidium iodide or 4′,6-diamidino-2-phenylindole (DAPI).

## Proliferation and migration assays

Cell proliferation was assessed by measuring metabolic activity using Cell Counting Kit-8 (Dojindo Laboratories, Kumamoto, Japan) according to the manufacturer's instructions. Transwell migration assays were conducted using ThinCert cell culture inserts with 8 μm pore diameter (Greiner Bio-One, Wemmel, Belgium) according to the manufacturer's instructions.

## Northern blotting, RT-PCR, real-time RT-PCR, and cDNA microarray analysis

Total RNA was isolated from Ca9-22 and HSC3 cells using NucleoSpin (Macherey-Nagel, Düren, Germany). Northern blot analysis was conducted using DIG Easy Hyb and an RNA

probe made with Dig RNA labeling Mixture (Roche Diagnostics, Basel, Switzerland) according to the manufacturer's protocol. RNA was reverse-transcribed to cDNA using oligo(dT) primers and M-MuLV reverse transcriptase (Thermo Fisher Scientific). PCRs were run with PrimeSTAR GXL DNA Polymerase (Takara) using the following thermal cycling program: 98°C for 1 min, 30 cycles of 98°C for 10 s, 58°C for 15 s, and 68°C for 20 s, and 68°C for 2 min. *GAPDH* was used for normalization. Quantitative RT-PCR was conducted using Platinum SYBR Green qPCR SuperMix-UDG (Thermo Fisher Scientific) and a LightCycler Nano system (Roche Life Science) using the following thermal cycling program: 95°C for 10 min, 40 cycles of 95°C for 10 s, 60°C for 10 s, and 72°C for 15 s. Primer sequences are available in S1 Table. For cDNA expression microarray analysis using SurePrint G3 Human GE 8x60K Ver.2.0 (Agilent Technologies, Santa Clara, CA, USA), total RNA was sent to a commercial service (Hokkaido System Science, Sapporo, Japan).

## Western blot analysis

Cells were lysed in RIPA buffer (20 mM Tris pH 7.4, 150 mM NaCl, 1% NP40, 0.1% SDS and 0.5% sodium deoxycholate, containing protease inhibitor cocktail cOmplete (Sigma-Aldrich). Protein samples were subjected to 10% SDS-PAGE and transferred to Hybond-ECL (GE Healthcare, Pittsburgh, PA, USA). The blotted membranes were blocked in 2% non-fat milk for 30 min and then incubated with primary antibodies (1:2,000) at 4°C overnight, followed by incubation with secondary antibodies (1:20,000) at room temperature for 1 h, and visualized using ECL Select (GE Healthcare).

## OSCC tissue specimens and immunohistochemistry

The study was reviewed and approved by the institutional ethics committee (registration number: D2012-078). As we used archival tissue specimens obtained for pathology diagnosis, the ethics committee approved waiver of the informed consent requirement in accordance with Ethical Guidelines for Clinical Studies by the Ministry of Health, Labor and Welfare of Japan. OSCC tissue specimens were randomly collected from sixty lateral tongue cancer cases from the archives of the Dental Hospital of Tokyo Medical and Dental University. Three tissue microarrays harboring 20 spots of 4 mm in diameter were constructed as previously described [17]. Fifty-one spots were fully evaluable after the loss of spots due to multiple sectioning. Histological evaluation was done as previously described [18]. Cancer-associated inflammation was scored based on the density of inflammatory cell infiltrates around tumor nests at the invasion front. Expression of AIRE, ICAM1, pSTAT1 and STAT1 in cancer was scored based on staining intensity in cancer versus normal epithelium, with "+/−" indicating that expression was undetected or detected but without noticeable upregulation in cancer, "+" indicating slight upregulation in cancer as revealed by a noticeable increase in staining intensity in cancer cells, and "++" indicating distinctive upregulation in cancer. All scorings were performed by two pathologists in a blinded manner, and the judgment was discussed until consent was obtained.

## Immunoprecipitation and chromatin immunoprecipitation (ChIP)

Cells were crosslinked with 1% formaldehyde at room temperature for 10 min. The reaction was stopped by incubating the cells with 10% glycine for 15 min. The cells were rinsed twice with PBS and lysed in RIPA buffer. The cell lysate was sonicated for 5 min and centrifuged to pellet the debris. The cleared lysate was divided into two parts, one of which was used for immunoprecipitation and western blot analysis, and the other for immunoprecipitation and DNA quantitative PCR assay. For immunoprecipitation and western blotting, the lysate was incubated with anti-ETS1 antibody (1:100 dilution) at 4°C overnight. Dynabeads Protein G

(Thermo Fisher Scientific) were added to the sample, which was incubated for an additional 1 h. The beads were washed with RIPA buffer three times using a magnet stand. Protein was eluted and crosslinks were reversed by heating the beads in SDS gel loading buffer (50 mM Tris-HCl [pH, 6.8], 2% SDS, 10% glycerol, 5% beta-mercaptoethanol, 0.01% bromophenol blue) at 95˚C for 10 min. Eluted proteins were subjected to western blot analysis. For ChIP, the lysate was incubated with anti-AIRE antibody at 4˚C overnight, then with Dynabeads Protein G for an additional 1 h. The sample was washed first with low-salt wash buffer (20 mM Tris-HCl [pH, 8.0], 150 mM NaCl, 2 mM EDTA, 1% Triton X-100, 0.1% SDS), then with high-salt wash buffer (20 mM Tris-HCl [pH, 8.0], 500 mM NaCl, 2 mM EDTA, 1% Triton X-100, 0.1% SDS), and then with LiCl wash buffer (10 mM Tris-HCl [pH, 8.0], 0.25 M LiCl, 1 mM EDTA, 1% NP-40, 1% sodium deoxycholate). DNA was eluted in elution buffer (100 mM NaHCO$_3$, 1% SDS). Crosslinks were reversed by incubation at 65˚C overnight, in the presence of RNaseA. Protein in the sample was digested with proteinase K at 60˚C for 1 h, and DNA was purified by phenol/chloroform extraction. Primers for quantitative PCR analysis are available in S1 Table.

## Statistical analysis

Quantitative data are presented as the mean ± standard error of at least three replicated experiments. Means were compared using Student's *t*-test or the Mann-Whitney U test. For analysis of relative protein or gene expression data, the ratio t-test was used. $P < 0.05$ was considered statistically significant.

## Results

### AIRE is induced in chemically induced cancer of the upper digestive tract in mice

We first examined AIRE expression in the healthy mouse thymus by immunohistochemical staining. AIRE-expressing cells were detected specifically in some, but not all epithelioid cells in the medulla (Fig 1A). AIRE-positive cells were also positive for keratin 14 (KRT14), confirming that they were mTECs. AIRE localized in the nuclei in punctate regions, which were more clearly observed by fluorescence than by chromogenic detection (Fig 1A). This AIRE protein distribution was in agreement with previous reports [19,20], validating the immunohistochemical method for AIRE detection.

We induced tumorigenesis in the upper digestive tract by administering the carcinogen 4-NQO to mice via the drinking water. Multiple tumors developed in the esophagi of all 4-NQO-treated mice (n = 5), whereas no tumor developed in control mice (n = 5). The tumors were histologically confirmed to be SCC or squamous intraepithelial neoplasia. We examined AIRE expression in these esophageal tumors. Although barely detectable in normal epithelium, AIRE expression was distinctively observed in neoplastic cells (Fig 1B and 1C). Nuclear dots as seen in the mTECs were evident in neoplastic cells based on immunofluorescence staining, but they were fewer than in the mTECs (Fig 1A and 1B). In chromogenic detection, cytoplasmic staining was observed in addition to nuclear staining (Fig 1C), the significance of which was uncertain.

### AIRE is upregulated in human OSCC tissues and cell lines

We first examined AIRE expression in human SCC using 7 SCC cell lines (6 OSCCs and 1 skin SCC) and cultured primary keratinocytes as a normal reference. AIRE expression was examined by RT-PCR using an intron-spanning primer set targeting *AIRE*. *AIRE* mRNA was

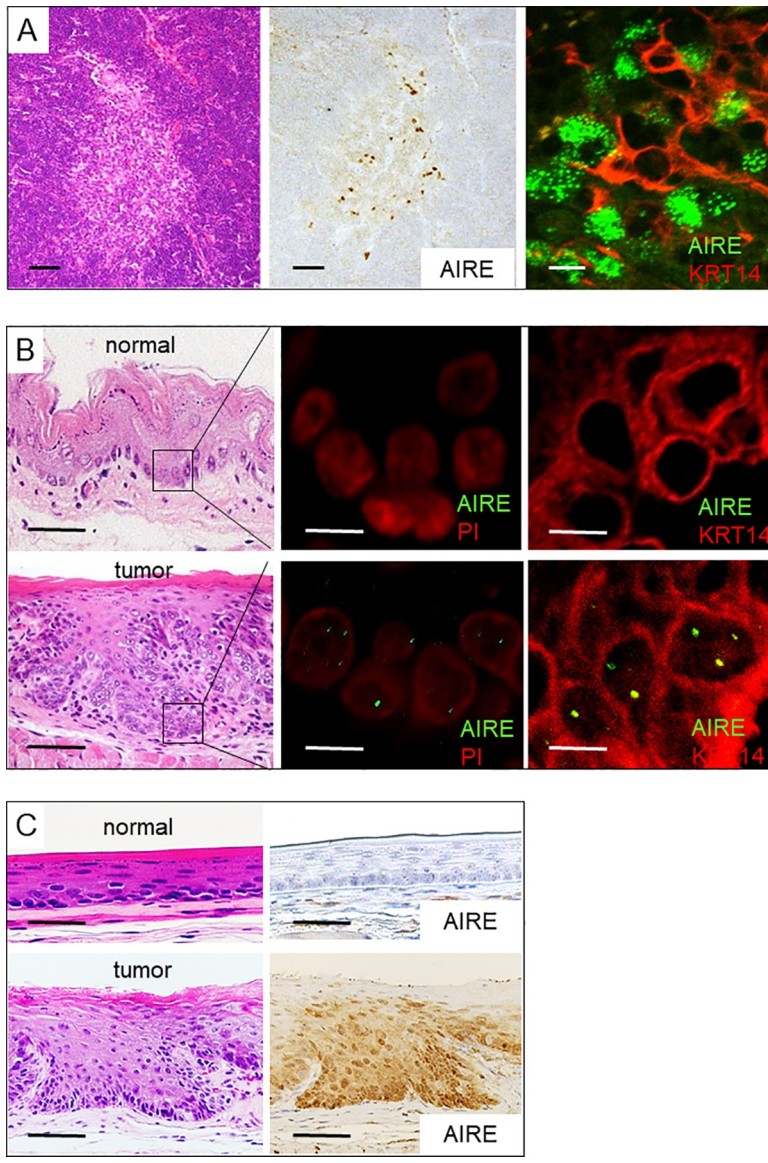

**Fig 1. AIRE expression in esophageal tumors in mice.** A) AIRE expression in the thymic medullary epithelium. Hematoxylin and eosin staining (left panel), immunohistochemical staining using 3,3'-DAB (middle panel), and immunofluorescence double staining with KRT14 (right panel). Scale bar: 50 μm (left and middle panels), 10 μm (right panel). B) AIRE expression in neoplastic cells of the esophagus. Immunofluorescence staining (middle panel) and immunofluorescence double staining with KRT14 (right panel). PI: propidium iodide. Scale bar: 50 μm (left), 5 μm (middle and right panels). C) Expression of AIRE in neoplastic cells of the esophagus as revealed by immunohistochemical staining using DAB. Scale bar: 50 μm. A representative negative-control image using nonspecific rabbit IgG instead of anti-AIRE antibody is shown in S1 Fig.

detected by RT-PCR in all the SCC cell lines examined as well as in normal keratinocytes. The level of expression was quantitated using real-time PCR analysis, which revealed that AIRE was upregulated in the oral cancer cell lines when compared with the normal keratinocytes (Fig 2A). Northern blot analysis failed to detect *AIRE* in any cells, suggesting a low amount of transcripts.

Western blot analysis using an anti-AIRE antibody revealed that all the SCC cell lines examined expressed AIRE at substantially higher levels than did normal keratinocytes (Fig 2B). We

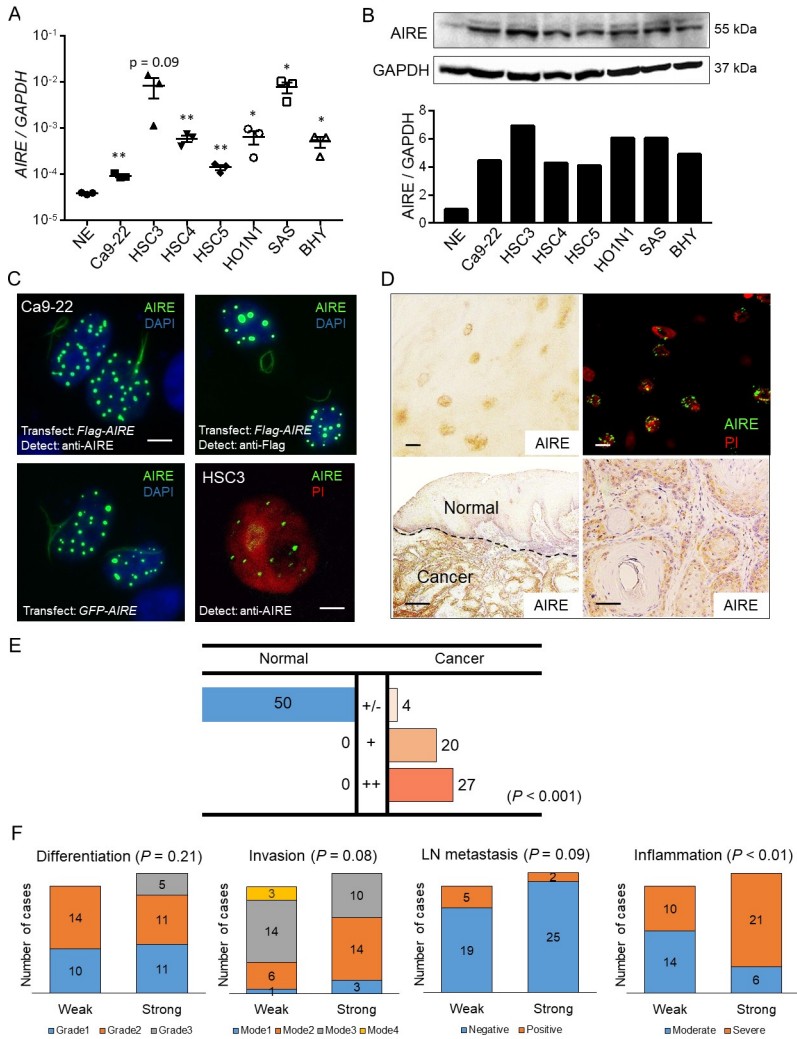

**Fig 2. AIRE expression in OSCC.** A) Real-time RT-PCR analysis of various cancer cell lines. NE: normal epithelium from gingiva. *GAPDH* was used for normalization. Data are shown on a logarithmic scale as the mean ± SEM of technical triplicates. * $P < 0.05$, ** $P < 0.01$ by Student's *t*-test. B) Upper panel: Western blot analysis of cancer cell lysates using anti-AIRE antibody. GAPDH was used as a loading control. Lower panel: Densitometric quantification of the western blot data. NE: normal epithelial primary cultured cells derived from neonatal foreskin. C) Nuclear localization of AIRE in cultured cells. Upper left and right panels: Ca9-22 cells were transfected with *Flag-AIRE*. The cells were fixed and subjected to immunofluorescence staining. Lower left panel: Ca9-22 cells were transfected with *GFP-AIRE*. The cells were examined under ultraviolet light. Lower right panel: endogenous AIRE in the nucleus of an HSC3 cell as revealed by immunofluorescence staining using laser scanning microscopy. PI: propidium iodide. Scale bar: 10 μm in upper left panel, 5 μm in lower right panel, scale bars were omitted in upper right and lower left panels. D) Representative images of immunohistochemical staining of AIRE in OSCC tissue specimens. Upper left, lower left, lower right panels: chromogenic detection using DAB. A representative negative-control image using nonspecific mouse IgG instead of anti-AIRE antibody is shown in S2 Fig. Upper right panel: fluorescence detection. Scale bar: 10 μm (upper left, upper right), 500 μm (lower left), 50 μm (lower right). E) Summary of immunohistochemical evaluation of AIRE in OSCC tissue specimens. Expression was compared between normal squamous epithelium and cancer in the same specimen and was scored as +/–: no upregulation in cancer; +: upregulation in cancer; ++: strong upregulation in cancer. Mann-Whitney U test. F) Comparison of AIRE expression with clinicopathological parameters (Mann-Whitney U test). "Weak" refers to cases with the score +/–or + in E), "strong" refers to the cases with score ++ in E).

further investigated the cellular localization of AIRE. For validation of the antibody in cultured-cell staining, Ca9-22 cells were transfected with *Flag-AIRE* and then stained using anti-

AIRE or anti-FLAG antibody 48 h after transfection. Both antibodies produced identical nuclear-dot staining patterns (Fig 2C). The same nuclear localization pattern was observed in live cells expressing GFP-AIRE, and protein distribution was not altered by fixation. Although endogenous AIRE expression was weaker than exogenous expression and the size and number of nuclear dots were lower than those in transfected cells, nuclear dots in HSC3 cells were positive for AIRE as indicated by laser scanning microscopy (Fig 2C). Next, we examined AIRE expression in 51 OSCC tissue specimens. AIRE was detected by immunohistochemical staining in the nuclei and cytoplasm of cancer cells. Immunofluorescence staining also revealed a nuclear dot pattern of AIRE, with substantially less cytoplasmic signal (Fig 2D). Therefore, we regarded the cytoplasmic staining as nonspecific and scored AIRE expression in OSCC based on the nuclear staining intensity in comparison to normal epithelium. Twenty-seven cases showed a distinctive increase ("++"), 20 cases showed a weak increase ("+"), and four cases showed unremarkable change ("+/−") in AIRE expression. These results indicated that AIRE expression is significantly increased in OSCC compared to normal epithelium (Fig 2E). Next, we examined whether AIRE expression was associated with clinicopathological parameters, including the differentiation grade, invasion pattern, lymph node metastasis, and level of inflammatory cell infiltration in the cancer stroma. OSCC with strong AIRE expression tended to exhibit severer inflammation (Fig 2F).

## AIRE promotes the expression of STAT1, ICAM1, CXCL10, CXCL11, and MMP9

To investigate the effect of AIRE on other genes, we performed cDNA microarray analysis to screen for genes upregulated by AIRE. Genes whose expression was increased by more than 2-fold in AIRE-expressing compared to control 293A cells in at least three microarray spots are listed in Fig 3A. Among them, we focused on *STAT1* and *ICAM1* because these genes have been detected at considerable levels in our previous cDNA microarray analysis of HSC3 cells [21], and reportedly are overexpressed and play important roles in OSCC [22,23]. Western blot analysis confirmed the upregulation of STAT1 and ICAM1 in AIRE-expressing clones (Fig 3B). Phosphorylated STAT1 increased in proportion to STAT1 expression.

To further explore the role of AIRE in OSCC, we disrupted AIRE expression in HSC3 cells. First, we tried transient transfection of siRNA, but the knockdown efficiency at the protein level was less than 20%. Therefore, we used the CRISPR-Cas9 system for gene knockout, and we established three independent *AIRE*-knockout HSC3 cell clones (HSC3/AIRE-C1, C2, and C3, Fig 4A). There was no significant difference in cell proliferation between HSC3/AIRE cells and HSC3 control cells (Fig 4B). A Transwell migration assay revealed that the migration rate was significantly suppressed in HSC3/AIRE-C1 and C2 when compared to control cells (Fig 4C). HSC3/AIRE-C3 exhibited a lower rate of migration, although the difference was not statistically significant. Western blot analysis revealed that ICAM1, STAT1, and phosphorylated STAT1 (pSTAT1) expression were significantly decreased in HSC3/AIRE as compared to control cells (Fig 4A).

To confirm these observations, we investigated the expression of ICAM1, and pSTAT1 in 51 OSCC tissue specimens by immunohistochemistry. In all cases examined, ICAM1, and pSTAT1 expression was increased in OSCC tissue compared to normal epithelium (Fig 4E). These findings supported the results of cell-culture experiments showing that AIRE promoted STAT1, and ICAM1 expression. AIRE stimulates the expression of the cancer-related proinflammatory genes *MMP9*, *CXCL10*, and *CXCL11* [6]. In line herewith, we found significantly reduced expression of *MMP9*, *CXCL10*, and *CXCL11* in HSC3/AIRE⁻ cells (Fig 5A), and significant upregulation of *MMP9*, *CXCL10*, and *CXCL11* in AIRE-expressing 293A cells (Fig 5B).

A

| Gene Name | Gene bank | Description | Fold change | P Value |
|-----------|-----------|-------------|-------------|---------|
| FMOD | NM_002023 | Fibromodulin | 8.72 | 4.17E-07 |
| CDH1 | NM_004360 | Cadherin 1 | 7.21 | 1.82E-05 |
| GPM6A | NM_201591 | Glycoprotein m6a | 6.16 | 1.72E-07 |
| SLC40A1 | NM_014585 | Solute carrier family 40 member 1 | 6.13 | 1.76E-16 |
| PLA2G4A | NM_024420 | Phospholipase a2, group iva | 4.43 | 2.26E-10 |
| ICAM1 | NM_000201 | Intercellular adhesion molecule 1 | 4.32 | 3.76E-04 |
| LOX | NM_002317 | Lysyl oxidase | 3.38 | 3.42E-09 |
| STAT1 | NM_139266 | Signal transducer and activator of transcription 1 | 3.06 | 3.09E-10 |
| DUSP1 | NM_004417 | Dual specificity phosphatase 1 | 2.69 | 3.81E-09 |
| VWF | NM_000552 | Von willebrand factor | 2.42 | 2.64E-07 |
| JDP2 | NM_130469 | Jun dimerization protein 2 | 2.22 | 5.59E-07 |
| FABP5 | NM_001444 | Fatty acid binding protein 5 | 2.21 | 6.39E-07 |
| ITGAV | NM_002210 | Integrin subunit alpha v | 2.19 | 7.93E-07 |
| FZD5 | NM_003468 | Frizzled class receptor 5 | 2.18 | 9.13E-07 |

B

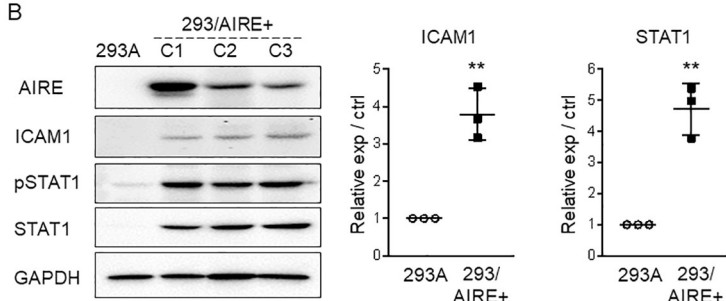

**Fig 3. AIRE promotes the expression of ICAM1 and STAT1 in 293A cells.** A) Genes that were upregulated more than 2-fold in in 293/AIRE+ (C-3) when compared to non-transfected 293A cells in at least three spots on the cDNA microarray. B) Left panel: Western blot analysis of ICAM1 and STAT1 in 293/AIRE+ cells. Right panel: Densitometric quantification of the western blot data. ** $P < 0.01$, by ratio t test.

These results suggested a common mechanism that regulates the expression of these cancer-related genes.

## AIRE promotes cancer-related gene expression in cooperation with ETS1

Nuclear dots containing AIRE reportedly are different from nuclear dots containing Sp100, the closest homologue of AIRE. However, we found that AIRE and Sp100 colocalized in nuclear dots as revealed by double staining for AIRE and Sp100 in *Flag-AIRE*-transfected cells (S3 Fig). Sp100 physically interacts with ETS1 and mediates its transcriptional activity [24,25]. ETS1 is a member of the ETS family of transcription factors that physically and functionally interact with numerous molecules to control transcription [26]. Furthermore, various genes that we found to be positively correlated with AIRE in terms of expression (i.e., *STAT1*, *ICAM1*, *MMP9*, *CXCL10*, and *CXCL11*) have been previously reported as remarkably upregulated in cDNA microarray analysis of epidermal keratinocytes from *Ets1* transgenic mice [27]. Therefore, we hypothesized that AIRE and ETS1 cooperatively contribute to expression regulation of these genes. The SCC lines expressed ETS1, along with STAT1 and ICAM1, at various levels (S4 Fig). We transiently transfected *Flag-AIRE* or/and *Ets1* into 293A cells, which do not express ETS1 at a detectable level as indicated by western blot analysis, and we examined the expression of the aforementioned genes. Western blot analysis revealed increases in STAT1 and ICAM1 expression upon overexpression of AIRE or/and ETS1 (Fig 6A and 6B). Coexpression of AIRE and ETS1 resulted in enhanced expression of STAT1 and ICAM1. Quantitative

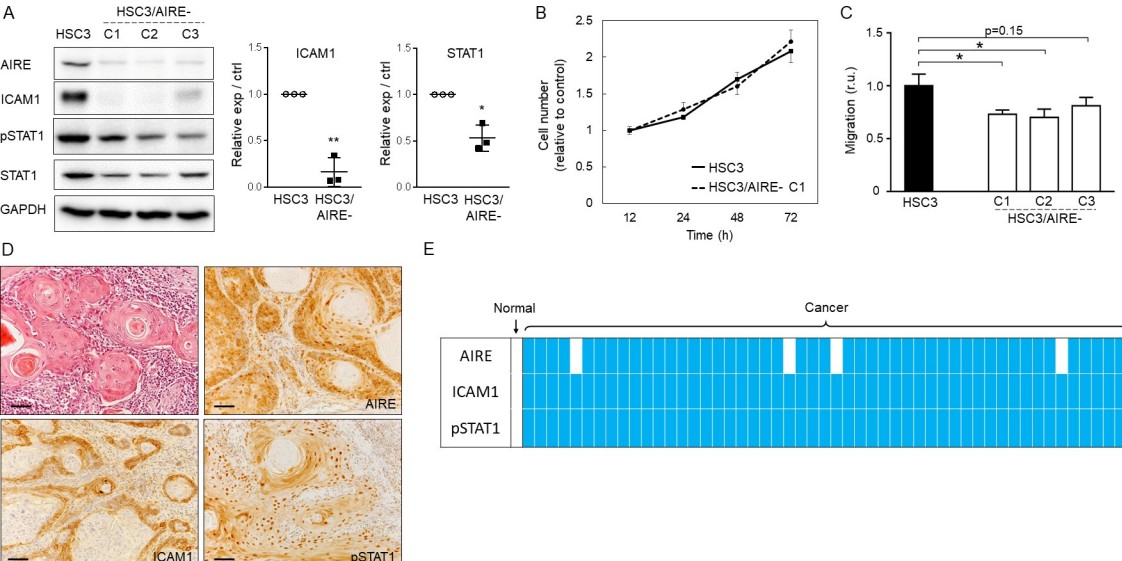

**Fig 4. STAT1, and ICAM1 are upregulated in OSCC.** A) Left panel: Western blot analysis of ICAM1, pSTAT1, and STAT1 in HSC3/AIRE⁻clones (C1, C2, C3). GAPDH was used as a loading control. Right panel: Densitometric quantification of the western blot data. * $P < 0.05$; ** $P < 0.01$, by ratio t test. B) Cell proliferation of HSC3 and HSC3/AIRE⁻C1. Data are shown as the mean + SEM of biological triplicates. C2 and C3 also showed similar proliferation rates as HSC3. Student's *t*-test. C) Transwell migration assay. Data are shown as the mean + SEM of biological triplicates. *$P < 0.05$, by Student's *t*-test. D) Representative images of immunohistochemical staining. Scale bar: 50 μm. Representative negative-control images are shown in S2E Fig) Schematic summary of immunohistochemical expression or AIRE, ICAM1, and pSTAT1 in 51 cases of OSCC. Horizontal grids represent the cases. Filled squares denote upregulation compared to normal epithelium in the same specimen. Blank squares denote no upregulation, i.e., similar staining intensity as in normal epithelium.

RT-PCR revealed that *CXCL10*, *CLCL11*, and *MMP9* expression was upregulated by ETS1 expression, and even more so upon cotransfection with *Flag-AIRE* (Fig 6C).

To examine the physical interaction between AIRE and ETS1, we cotransfected *Flag-AIRE* or/and *Ets1* into 293A cells and used these cells for immunoprecipitation. Conventional immunoprecipitation using anti-ETS1 antibody did not show coprecipitation of AIRE with ETS1, which indicates that their interaction is weak or their functional cooperation is not mediated through direct physical interaction. ETS1 and AIRE may be recruited to nearby chromatin regions. To check this possibility, we performed a modified immunoprecipitation assay. Formaldehyde crosslinking was performed before cell lysis, and the DNA was sheared by sonication. Immunoprecipitation was conducted using the anti-ETS1 antibody, and recovered protein was examined by western blot analysis. AIRE coprecipitated with ETS1 (Fig 6D). We next performed conventional ChIP assays to assess whether AIRE was recruited to the promoters of the aforementioned genes. ChIP-quantitative PCR revealed that promoter fragments of *STAT1*, *ICAM1*, *CXCL10*, and *MMP9* were enriched in the AIRE precipitate (Fig 6E). These results implicate that AIRE promotes cancer-related gene expression in cooperation with ETS1.

## Discussion

Although accumulating evidence suggests that AIRE is expressed in extrathymic tissues, there are remarkable discrepancies across studies, depending on the techniques used. *AIRE* mRNA has been detected by RT-PCR in lymph nodes, tonsils, gut-associated lymphoid tissues, spleen, liver, testes, and ovaries [28]. Thymic *AIRE* mRNA expression is the highest and could also be detected by northern blotting [29,30]. As studies have concentrated mainly on lymphoid

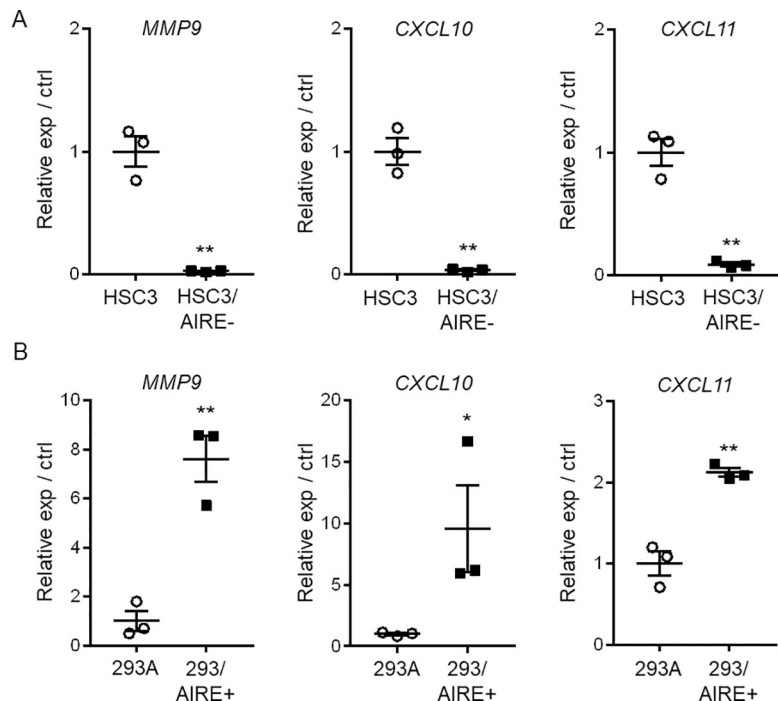

**Fig 5. Relative expression of proinflammatory genes.** Gene expression in A) HSC3/AIRE⁻, and B) 293/AIRE+ in comparison to respective non-transformed cells as measured by real-time RT-PCR. *GAPDH* was used for normalization. Data are shown as the mean + SEM of technical triplicates and are representative of three independent experiments. *$P < 0.05$, ** $P < 0.01$, by Student's *t*-test.

tissues, the skin and upper digestive tract mucosa have been left unexplored. Only a few studies have reported AIRE expression in epithelial tissues. *In-situ* hybridization and immunohistochemistry failed to detect AIRE expression in human skin [8], whereas *AIRE* expression has been detected by RT-PCR in HaCaT keratinocytes [11], and in A431 cells treated with 12-O-tetradecanoylphorbol 13-acetate (TPA) [6]. TPA is an activator of the protein kinase C pathway, which induces inflammation, and acts as a strong carcinogen when topically applied to skin. *Aire* mRNA expression was barely detectable in normal mouse skin by *in-situ* hybridization, but was induced in the epidermis of TPA-treated mouse ear and genetically induced mouse skin tumor. Still, induced epidermal *Aire* expression was substantially lower than that in the thymus [6]. Our results showing that AIRE is expressed in OSCC are consistent with these previous observations, and suggest that AIRE is induced in the epithelium under pathological conditions.

We examined the expression of several tissue-specific antigens, including PTH, INS, GH1, which are induced in mTECs in an AIRE-dependent manner [7], in OSCC cell lines. None of these were detected in the OSCC cell lines examined or induced in AIRE-overexpressing cells. This is plausible considering the presumptive function of AIRE in reinstigating suspended transcription. Instead, we found evidence of upregulation of STAT1, and ICAM1 by AIRE. The coordinated overexpression of STAT1, and ICAM1 in OSCC is consistent with findings in previous studies [23,28,31–33]. Close correlations between STAT1, and ICAM1 expression in epithelial cells have been demonstrated in previous studies. Interferon gamma induces ICAM1 expression through activation of STAT1 [34–36]. Although the regulatory mechanism of STAT1 transcription is unclear, our finding that AIRE induces STAT1 expression suggests that AIRE may increase the expression of ICAM1 through STAT1 upregulation.

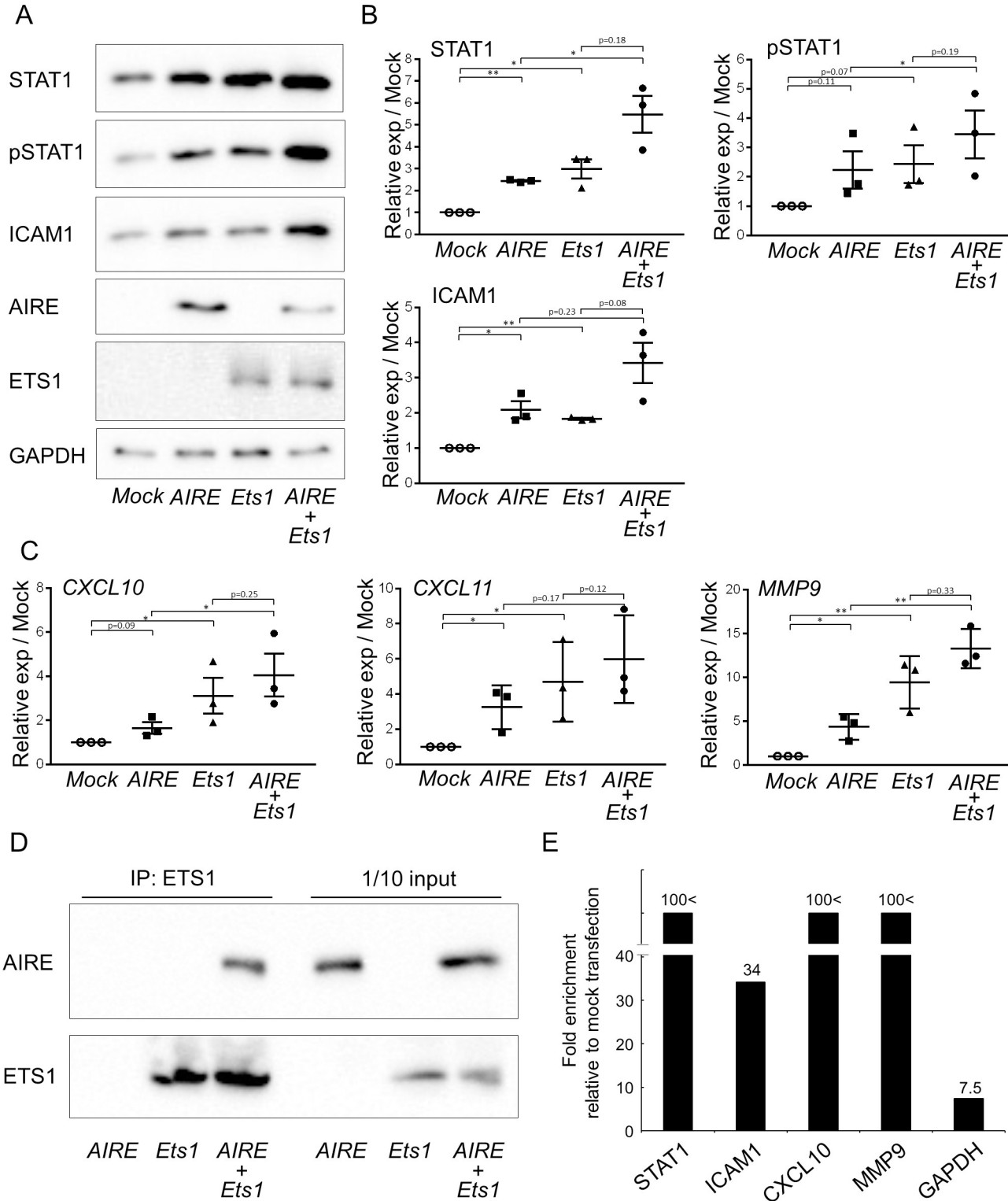

**Fig 6. Physical and functional interaction of AIRE and ETS1.** A) Expression of STAT1, pSTAT1, and ICAM1 in 293A cells 48 h after transfection with *Flag-AIRE* or/and *ETS1*, or mock transfection. The blot shown is representative of three independent experiments. B) Densitometric analysis of data in A). Data are shown as the mean ± SEM of technical triplicates and are representative of three independent experiments. $^*P < 0.05$, $^{**}P < 0.01$, by ratio t-test. C) Relative gene expression of *CXCL10*, *CXCL11*, and *MMP9* in comparison to mock transfection as revealed by real-time PCR. Data are shown as the mean ± SEM of triplicate wells and are representative of two independent experiments. $^* P < 0.05$, $^{**} P < 0.01$, by ratio t-test. D) Immunoprecipitation and western blot analysis of ETS1 and AIRE. Formaldehyde crosslinking was performed prior to lysis. The lysates were sonicated to shear the DNA.

Immunoprecipitation with the anti-ETS1 antibody was performed, and recovered protein was examined by western blot analysis. One tenth of the sample used for immunoprecipitation was loaded as a control. E) ChIP assay of 293A cells transiently transfected with *Flag-AIRE*. Enrichment of promoter fragments was measured by real-time PCR. Values were standardized to the input, and then to mock transfection. Relative values higher than 100 are indicated as 100< and a scale break was used in the Y-axis to allow intuitive interpretation of the relative expression. *GAPDH* was used as a reference. Data are representative of three independent experiments.

We found that AIRE stimulates *MMP9*, *CXCL10*, and *CXCL11*, which is consistent with findings by Hobbs and colleagues [6]. These authors demonstrated that AIRE and KRT17 are recruited to a specific region in the promoters of these proinflammatory genes carrying an NF-κB consensus motif [6]. Interestingly, the same spectrum of genes whose expression was dependent on AIRE in keratinocytes under stress in our study, including *Stat1*, *Icam1*, *Mmp9*, *Cxcl10*, and *Cxcl11*, was reportedly induced in keratinocytes of *Ets1* transgenic mice [27]. Microarray analysis of *Ets1* transgenic in comparison with wild-type mouse epidermis revealed the upregulation of *Stat1* (2.4-fold), *Icam1* (2.8-fold), *Mmp9* (8.9-fold), *Cxcl10* (8.2-fold), and *Cxcl11* (3.3-fold) in the former [27]. Unfortunately, data on *Aire* expression were lacking in this previous microarray study, but the results in this and previous studies suggest that AIRE belongs to this group of genes that are induced in epithelial cells under stress condition.

AIRE regulates the transcription of numerous divergent genes whose expression is driven by various promoters and transcription factors. This suggests that AIRE is a broad transcriptional activator, rather than a specific DNA-binding transcription factor. The mechanism underlying this promiscuous transcriptional activation is not fully understood, but it has been proposed that AIRE is recruited to RNA polymerase II-rich regions where transcription of AIRE-responsive genes is initiated but interrupted, and AIRE releases the stalled RNA polymerase II complex to resume transcription [37–39]. Further, AIRE has been suggested to be preferentially recruited to so-called super-enhancers; genomic regions rich in enhancers and transcriptional regulators [40]. Through its global transcriptional activation, AIRE might assist cancer-associated gene expression and thus promote the cancer phenotype.

The regulatory mechanism of AIRE expression has not been clearly understood. The *AIRE* minimal promoter region contains binding domains for Sp1, AP-1, NF-Y [41], and Ets transcription factors [42]. Indeed, ETS1 and ETS2 enhanced TPA-induced *AIRE* promoter activity in HeLa cells [42]. Furthermore, *AIRE* expression is regulated by KRT17 in cooperation with the ribonucleoprotein hnRNPK [6]. As KRT17 was robustly induced in all the oral cancer tissues examined [31], the induction of AIRE in OSCC may be attributed to KRT17-mediated regulation. Further research is needed to elucidate the details of the mechanism of AIRE induction in cancer cells.

In conclusion, we demonstrated a novel role of AIRE in extra-thymic tissue under pathologic condition. AIRE is upregulated in OSCC and promotes the expression of cancer-related genes, at least in part by functional interaction with ETS1.

## Supporting information

**S1 Table. Sequences of the PCR primers used in this study.**
(PDF)

**S1 Fig. Representative negative control image of immunohistochemical staining of mouse tissue.** Scale bar: 100 μm.
(PDF)

**S2 Fig. Representative negative control image of immunohistochemical staining of human OSCC.** Scale bar: 100 μm.
(PDF)

**S3 Fig. Sp100 and transfected AIRE in the nuclei of Ca9-22 cells as revealed by immunofluorescence staining and laser scanning microscopy.** There is a transfected cell in the center, in which AIRE colocalizes with endogenous Sp100. Note the Sp100 expression in the surrounding non-transfected cells. Scale bar: 5 μm.
(PDF)

**S4 Fig. Expression of STAT1, pSTAT1, ICAM1, and ETS1 in various OSCC cell lines.** Western blot analysis. GAPDH was used as a loading control.
(PDF)

**S1 File. Raw Images.**
(PDF)

## Acknowledgments

I would like to express my gratitude to my advisor, Kei Sakamoto, for his support, patience, and encouragement throughout my study in Japan. I am also grateful to all the people who contributed to the research. A very special word of thanks goes for my parents, who gave me the very best that they could.

## Author Contributions

**Conceptualization:** Chi Thi Kim Nguyen, Akira Yamaguchi, Kei Sakamoto.

**Data curation:** Chi Thi Kim Nguyen, Wanlada Sawangarun, Masita Mandasari, Kei Sakamoto.

**Formal analysis:** Chi Thi Kim Nguyen, Kei Sakamoto.

**Funding acquisition:** Akira Yamaguchi, Kei Sakamoto.

**Investigation:** Chi Thi Kim Nguyen, Wanlada Sawangarun, Masita Mandasari, Kei Sakamoto.

**Methodology:** Chi Thi Kim Nguyen, Kei Sakamoto.

**Project administration:** Chi Thi Kim Nguyen, Akira Yamaguchi, Kei Sakamoto.

**Resources:** Wanlada Sawangarun, Masita Mandasari, Kei-ichi Morita, Hiroyuki Harada, Kou Kayamori, Kei Sakamoto.

**Supervision:** Kei-ichi Morita, Kou Kayamori, Akira Yamaguchi, Kei Sakamoto.

**Validation:** Chi Thi Kim Nguyen, Kei Sakamoto.

**Visualization:** Chi Thi Kim Nguyen, Kei Sakamoto.

**Writing – original draft:** Chi Thi Kim Nguyen, Kei Sakamoto.

**Writing – review & editing:** Chi Thi Kim Nguyen, Kei Sakamoto.

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
