## [Decision Letter · Decision Letter 0]

6 Nov 2019

PONE-D-19-24858

AIRE is induced in oral squamous cell carcinoma and promotes cancer gene expression.

PLOS ONE

Dear Dr Sakamoto,

Thank you for submitting your manuscript to PLOS ONE. After careful consideration, we feel that it has merit but does not fully meet PLOS ONE’s publication criteria as it currently stands. Therefore, we invite you to submit a revised version of the manuscript that addresses the points raised during the review process.

We would appreciate receiving your revised manuscript by Dec 21 2019 11:59PM. To enhance the reproducibility of your results, we recommend that if applicable you deposit your laboratory protocols in protocols.io, where a protocol can be assigned its own identifier (DOI) such that it can be cited independently in the future. For instructions see: http://journals.plos.org/plosone/s/submission-guidelines#loc-laboratory-protocols

We look forward to receiving your revised manuscript.

Kind regards,

Roberto Mantovani

Academic Editor

PLOS ONE

Journal Requirements:

1. At this time, we request that you  please report additional details in your Methods section regarding animal care, as per our editorial guidelines.

(1) Please state the number of mice used in the study.

(2) Please state the source, product number and lot number of the 4-nitroquinoline-1-oxide (4-NQO) used in the study

Thank you for your attention to these requests.

2. Please provide additional information about each of the cell lines used in this work, including culture conditions (incubation temperature, CO2)  and any quality control testing procedures (authentication, characterisation, and mycoplasma testing). For more information, please see http://journals.plos.org/plosone/s/submission-guidelines#loc-cell-lines.

Reviewers' comments:

Reviewer's Responses to Questions

**Comments to the Author**

1. Is the manuscript technically sound, and do the data support the conclusions?

Reviewer #1: No

Reviewer #2: Partly

2. Has the statistical analysis been performed appropriately and rigorously? 

Reviewer #1: No

Reviewer #2: Yes

3. Have the authors made all data underlying the findings in their manuscript fully available?

Reviewer #1: No

Reviewer #2: Yes

4. Is the manuscript presented in an intelligible fashion and written in standard English?

Reviewer #1: Yes

Reviewer #2: Yes

5. Review Comments to the Author

Reviewer #1: Experiments and in particular PCR and WB should be quantitative and statically significant. Statistical evaluation of the data should be provided. As detailed in the attached comments controls for immustaining should be provided.

In general Figures are of low quality and do not meet the scientific criteria to be accepted for publication in PlosONE.

Reviewer #2: In the manuscript “AIRE is induced in oral squamous cell carcinoma and promotes cancer gene

expression” the authors performed several experiments to demonstrated that AIRE, a thymic transcription factor, is expressed also in extra-thymic tissue (skin) and is overexpressed in OSCC. Their results indicate that AIRE induces in these tumors the expression of cancer-related genes and suggest it works through a functional interaction with ETS1.

Although the issue is interest and relevant, some results are very preliminary.

Comments

-In fluorescence AIRE is very poorly expressed in the nucleus in comparison with normal cells but in IHC the nuclei seem very positive. How do you explain it?

-The authors state that in most cases, KRT17, ICAM1, and pSTAT1 expression was increased in OSCC tissue compared to normal epithelium. Are these the same cases in which AIRE is overexpressed? If not how can you reconcile the results?

-AIRE induced expression of KRT17. Why the ko of KRT17 leads to downregulation of AIRE? Which is the molecular mechanism trough which KRT17 regulates AIRE? Is KRT17 recruited to the AIRE promoter? If this result is shown in the manuscript more experiments addressing the mechanism should be performed.

-Which is the molecular mechanism leading to the fact that ETS1 was induced in stable 293/AIRE+ clones and decreased in HSC3/AIRE– cells? Is AIRE recruited to the ETS1 promoter, too? Is AIRE regulating ETS1 expression, too? Again as for KRT17 if this result is shown in the manuscript more experiments addressing the mechanism should be performed.

-The results of figure 8 are only observational and do not implicate at all that AIRE is recruited to the chromosomal regions where ETS1 initiated transcription. re-ChIP experiments with or without the two proteins should be performed to demonstrate this.

-The genes are written sometimes in italicus sometimes no in a casual manner (ie: not according with gene or protein).

6. PLOS authors have the option to publish the peer review history of their article (what does this mean?). If published, this will include your full peer review and any attached files.

Reviewer #1: No

Reviewer #2: Yes: Giulia Piaggio

---

## [Author Response · Author response to Decision Letter 0]

19 Dec 2019

Responses to the Reviewers’ comments

We thank the reviewers for the helpful suggestions. Changes made in the manuscript in response to the comments are highlighted in yellow.

Reviewer #1: 

Q1: Experiments and in particular PCR and WB should be quantitative and statically significant. Statistical evaluation of the data should be provided. As detailed in the attached comments controls for immustaining should be provided.

A1: We thank the reviewer for this valuable comment. Following the reviewer's instructions, relevant figures were substantially revised. PCR and western blot results were quantitated, and statistical analysis, when relevant, was performed. Control images for immunostaining were provided in S1 Fig and S2 Fig.

Q2: Data from Hobbs RP et al. Nat. Genetics 2015 doi: 10.1038/ng.3355 demonstrated that AIRE induction and nuclear localization strongly depends upon functional interaction between Krt17 and hnRNPK. In this work, instead, the authors describe induction of Krt17 by AIRE thus suggesting a direct role of AIRE transcriptional factor in Krt17 induction. Thus, a critical point should be to understand the relevance of Krt17 in AIRE-induced transcriptional pathway or the potential existence of a regulatory feedback loop between them, a relevant point that has not been addressed by the authors.

The work is potentially interesting however I have several concerns especially about the way the data have been done and presented.

A2: We appreciate the reviewer’s constructive suggestion. We started this study inspired by the close relationship between KRT17 and AIRE. Several data using cultured cells indicated that a change in KRT17 expression was associated with a change in AIRE expression, although the mechanism of KRT17 upregulation in the AIRE-overexpressing cells was yet to be elucidated. On the other hand, Hobbs et al. have already shown that AIRE expression is induced by the cooperative action of KRT17 and hnRNPK. Thus, the induction of AIRE in oral cancer tissue would be explained by KRT17-mediated regulation, because KRT17 was robustly induced in all the oral cancer tissue examined. This possibility was added to the Discussion.

Our cell culture data implied at least four types of gene induction relationships, as follows: AIRE >> ETS1, ETS1 >> KRT17, KRT17 >> AIRE, AIRE >> KRT17.

As the reviewer pointed out, additional experiments are needed to clarify these relationships in terms of the underlying molecular mechanisms. Following the reviewers’ suggestions, we performed additional experiments, where we transiently transfected AIRE and ETS1 into 293A cells and confirmed that KRT17 mRNA expression was increased by both. Likely, there is a complex correlation between these genes, with bidirectional feedback and links to other signaling cascades. Each of the above questions is a big subject, and as the reviewer pointed out, the impact of KRT17 on AIRE function is also an important issue. That said, it would be too complicated to address all these questions in this paper, and this would obscure the main topic of the article. The main topic is that AIRE is overexpressed in oral squamous cell carcinoma and increases the expression of genes related to cancer behavior. Although the induction of AIRE in oral cancer cells may be the consequence of KRT17 upregulation, the function of KRT17 is not explicitly relevant to this story. Therefore, we decided to delete the KRT17 data from the paper. Data on ETS1 expression in AIRE-overexpressing clones were also deleted. We believe that the revised manuscript is more concise and readable. The biological complexity underlying the phenomena might be underrepresented, but these issues should be examined in detail in future studies.

Q3: The author should test if Krt17 knock-out or depletion plays any effects on AIRE-dependent gene regulation. This should easily clarify the role of Krt17 in this pathway.

A3: We thank the reviewer for this valuable suggestion. As mentioned above, KRT17 results were deleted from the present paper for it to be more concise. We agree that the role of Krt17 in AIRE-dependent gene regulation is an important topic. As the cells available in our repository either express both KRT17 and AIRE, or lack both, co-transfection of KRT17 and AIRE into 293A cells is a versatile experimental model to examine this topic. There seems to be a complex bidirectional regulatory feedback between the cancer genes, which may make it difficult to determine whether the effect is due to direct functional interaction or a secondary one via other signaling cascades. We would like to examine these questions in a future study.

Q4: Experiments in Fig. 2A and B should be quantitated and statically evaluation of the data should be provided.

A4: We performed real-time PCR in triplicate, and quantitative data were added in the revised manuscript.

Q5: In Fig. 3A, a panel showing mock transfected cells is missing. The experiment in Fig. 3C is not quantitative and should be repeated by quantitative PCR with appropriate internal controls.

A5: Real-time PCR was performed to evaluate KRT17 expression in AIRE-transfected 293A cells, which confirmed the upregulation. However, we chose not to include data related to the interaction between AIRE and KRT17 in this manuscript, and thus, Figure 3 was deleted.

Q6: In Fig. 3D the authors should explain why expression of Krt17 in clone C1 is so low given that AIRE is abundantly expressed compared to the other clones. Again, this experiment is not quantitative and statistics is not present.

A6: We thank the reviewer for this constructive comment. The level of expression of KRT17 and other genes did not seem to consistently parallel that of AIRE in the stable clones. We have frequently experienced such clone variation, which may be due to differences in integration sites.

Q7: In Fig. 4B, STAT1 levels appear to be comparable despite the different level of AIRE in C1, C2 and C3 clones.

A7: Such clone variation may be due to differences in integration sites. We assume that, as we did not identify the integration site or the epigenetic states of each clone, comparison between clones is not very informative. Therefore, we made comparisons between the non-transformed parent cells and the group of clones. 

Q8: The WB presented in Fig. 4B is not quantitated.

A8: Densitometry results were added (Fig. 3A).

Q9: The WB in Fig. 7 should be quantitated, the AIRE panel in Fig. 7B

A9: This is overexpressed AIRE, and this is not an experiment to check the dose-dependency. Densitometry of AIRE and ETS1 is possible, but seems irrelevant in the context. The expression of other genes was quantitated and statistical analysis was performed (Fig. 6B, C).

Reviewer #2: 

Q10: In fluorescence AIRE is very poorly expressed in the nucleus in comparison with normal cells but in IHC the nuclei seem very positive. How do you explain it?

A10: Immunohistochemical staining employs a signal enhancing technique, which can dramatically increase the staining intensity. However, intracellular localization tends to become blurred. As seen in Fig 1A, the dot pattern in TEC cells was clearly visible by immunofluorescence, whereas the nuclei were broadly stained by chromogenic substrate. In cancer cells, AIRE expression was weaker than the physiological expression in normal TEC cells. To obtain sufficient signal, we used a color development time of 5–10 min, while monitoring the color intensity. Longer incubation can increase non-specific staining, but no significant staining was observed in the control specimen in which the primary antibody was replaced with non-specific IgG. 

Q11: The authors state that in most cases, KRT17, ICAM1, and pSTAT1 expression was increased in OSCC tissue compared to normal epithelium. Are these the same cases in which AIRE is overexpressed? If not how can you reconcile the results?

A11: We thank the reviewer for this constructive comment. KRT17, ICAM1, and pSTAT1 were increased in all the OSCC cases examined. As STAT1 staining was much weaker than that of pSTAT1, apparently because the STAT1 antibody is less optimal for immunohistochemical staining on formalin-fixed paraffin embedded tissue specimens, upregulation of STAT1 was not confirmed in all the cases. To avoid confusion, we decided to delete immunohistochemical results for STAT1. Overall, AIRE upregulation in OSCC was detected in 47/51 cases, and KRT17, ICAM1, pSTAT1 were upregulated in 51/51 cases. Thus, their co-upregulation is a common feature in OSCC. 

Q12: AIRE induced expression of KRT17. Why the ko of KRT17 leads to downregulation of AIRE? Which is the molecular mechanism trough which KRT17 regulates AIRE? Is KRT17 recruited to the AIRE promoter? If this result is shown in the manuscript more experiments addressing the mechanism should be performed.

A12: We thank the reviewer for this constructive comment. AIRE downregulation in KRT17-KO cells is consistent with the data from Hobbs et al. that clearly demonstrated that AIRE is induced by Krt17 in association with hnRNPK. As mentioned above in response to a comment by reviewer #1, we considered that data related to KRT17 are better to be further analyzed in a future study. Therefore, data on KRT17 were removed from the manuscript.

Q13: Which is the molecular mechanism leading to the fact that ETS1 was induced in stable 293/AIRE+ clones and decreased in HSC3/AIRE– cells? Is AIRE recruited to the ETS1 promoter, too? Is AIRE regulating ETS1 expression, too? Again as for KRT17 if this result is shown in the manuscript more experiments addressing the mechanism should be performed.

A13: We thought that ETS1 upregulation in the AIRE-overexpressing clones is an interesting phenomenon that should be shown as additional information. Maybe there is a complex multidirectional feedback mechanism that allows the cells to adapt to a new state of balanced gene expression. However, this information might confuse the readers, and more experiments are definitely needed to understand the mechanisms underlying this finding. Therefore, we decided not to present this result in the current paper. The possibility that AIRE promotes ETS1 expression is to be examined in a future study. 

Q14: The results of figure 8 are only observational and do not implicate at all that AIRE is recruited to the chromosomal regions where ETS1 initiated transcription. re-ChIP experiments with or without the two proteins should be performed to demonstrate this.

A14: Although the IP result suggested that ETS1 and AIRE exist as a complex, we agree that it is an overstatement that AIRE is recruited to the chromosomal regions where ETS1 initiates transcription. We attempted ChIP-re-ChIP experiments; however, unfortunately, they failed to provide direct evidence supporting the above hypothesis so far. We re-organized the current manuscript and decided to study the detailed mechanism in a future study. Accordingly, we deleted the data in the original Fig 8, and revised the manuscript to avoid any overstatements. 

Q15: The genes are written sometimes in italicus sometimes no in a casual manner (ie: not according with gene or protein).

A15: We apologize for this mistake. The notation of genes and proteins has been carefully revised.

---

## [Decision Letter · Decision Letter 1]

3 Jan 2020

AIRE is induced in oral squamous cell carcinoma and promotes cancer gene expression.

PONE-D-19-24858R1

Dear Dr. Sakamoto,

We are pleased to inform you that your manuscript has been judged scientifically suitable for publication and will be formally accepted for publication once it complies with all outstanding technical requirements.

With kind regards,

Roberto Mantovani

Academic Editor

PLOS ONE

Additional Editor Comments (optional):

Reviewers' comments:

Reviewer's Responses to Questions

**Comments to the Author**

1. If the authors have adequately addressed your comments raised in a previous round of review and you feel that this manuscript is now acceptable for publication, you may indicate that here to bypass the “Comments to the Author” section, enter your conflict of interest statement in the “Confidential to Editor” section, and submit your "Accept" recommendation.

Reviewer #1: All comments have been addressed

Reviewer #2: All comments have been addressed

2. Is the manuscript technically sound, and do the data support the conclusions?

Reviewer #1: Yes

Reviewer #2: Yes

3. Has the statistical analysis been performed appropriately and rigorously? 

Reviewer #1: Yes

Reviewer #2: Yes

4. Have the authors made all data underlying the findings in their manuscript fully available?

Reviewer #1: Yes

Reviewer #2: Yes

5. Is the manuscript presented in an intelligible fashion and written in standard English?

Reviewer #1: Yes

Reviewer #2: Yes

6. Review Comments to the Author

Reviewer #1: Regrettably, the authors did not add all the informations requested by this reviewer. However, they decided to remove data from the previous version of the manuscript to make it more focused. Moreover, they add all the quantifications data requested. Therefore I think that the manuscript as it is in the revised version is acceptable for publication.

Reviewer #2: In the first version of the manuscript there were several preliminary data that would need further experiments. In this revised version, the authors deleted those preliminary data. The paper is now more focused, although simplified. In line with the policy of PONE, it can be accepted for publication.

7. PLOS authors have the option to publish the peer review history of their article (what does this mean?). If published, this will include your full peer review and any attached files.

Reviewer #1: Yes: Viola Calabrò

Reviewer #2: Yes: Giulia Piaggio

---

## [Editor Report · Acceptance letter]

9 Jan 2020

PONE-D-19-24858R1 

AIRE is induced in oral squamous cell carcinoma and promotes cancer gene expression. 

Dear Dr. Sakamoto:

I am pleased to inform you that your manuscript has been deemed suitable for publication in PLOS ONE. Congratulations! Your manuscript is now with our production department. 

With kind regards,

on behalf of

Prof. Roberto Mantovani 

Academic Editor

PLOS ONE